# Prevalence of Malnutrition among Syrian Refugee Children from Lebanon

**DOI:** 10.3390/life13020453

**Published:** 2023-02-06

**Authors:** Tamara Mroue, Betlem Heras, Jose M. Soriano, María Morales-Suarez-Varela

**Affiliations:** 1Observatory of Nutrition and Food Safety for Developing Countries, Food & Health Lab, Institute of Materials Science, University of Valencia, 46980 Paterna, Spain; 2Joint Research Unit on Endocrinology, Nutrition and Clinical Dietetics, Health Research Institute La Fe-University of Valencia, 46026 Valencia, Spain; 3Unit of Preventive Medicine and Public Health, Department of Preventive Medicine and Public Health, Food Sciences, Toxicology and Forensic Medicine, University of Valencia, 46100 Burjassot, Spain; 4CIBER in Epidemiology and Public Health (CIBERESP), Institute of Health Carlos III, 28029 Madrid, Spain

**Keywords:** anthropometrical parameters, children, health, nutritional status, Syrian refugee

## Abstract

Today, the situation of Syrian refugees is one of the world’s worst humanitarian crises. To estimate the prevalence of malnutrition among pediatric populations of Syrian refugees, 176 Syrian refugee children, with stays of more than two years at three refugee camps (Zalhé, Deddeh, and Kfar Jouz) or from the town of Yohmor, Lebanon were authorized by their parents to participate in this study. The children were anthropometrically evaluated and height-for-age Z-score (HAZ), weight-for-age Z-score (WAZ), and weight-for-height (WHZ) Z-scores were obtained and compared with WHO standards. Furthermore, mid-upper arm circumference (MUAC) was analyzed for screening children 6–59 months old. According to the anthropometric measures, no child met the criteria for chronic, global, acute malnutrition (CGAM), severe acute malnutrition (SAM), or moderate acute malnutrition (MAM). In the total sample, 49.4% of participants were moderately thin, with girls presenting a higher prevalence of thinness than that of boys. Thus, the absence of high rates of malnutrition was verified despite the magnitude of the Syrian refugee’s problem. The data provided by this study identify the need to carry out further research to assess anthropometric growth and nutritional status among long-staying refugee children in order to prevent any health issues that may arise in the future.

## 1. Introduction

The Syrian civil war has resulted in citizens and permanent residents of Syria who have now become new refugees, with this type of situation increasing in other parts of the world due to other wars. Data have suggested that the pre-war population of the Syrian Arab Republic was approximately 22 million, including permanent residents. The United Nations (UN) identified 13.5 million displaced persons who required humanitarian assistance. Often described as one of the largest refugee crises in history, and since early 2011, growing political instability and escalating violence in the region have caused Syrian citizens to seek asylum by fleeing to neighboring countries [1]. International observers consider that the human rights conditions in Syria have been exceptionally poor under the rule of the Arab Socialist Ba’ath Party–Syria Region (continuous since 1963) and, since 2008, have further deteriorated. According to the United Nations High Commissioner for Refugees, currently, there are approximately six million Syrian refugees residing mainly in Turkey, Lebanon, Jordan, Iraq, and Egypt [2]. In general, these forced displacements and subsequent life in refugee camps pose health risks for the displaced populations. Many of these refugees are trapped in camps with no possibility of returning to Syria or being taken in by other countries, leading to long stays in these camps. Resources are scarce, camps and settlements are overcrowded, and many are forced to sleep outdoors in freezing temperatures. The resources available on the ground are not adequate or sufficient; therefore, fast and effective aid is required [3] to protect and to promote adequate health and nutrition [4].

The World Health Organization (WHO) [5] has estimated that 34% of Syrian refugees reside in Lebanon, with 54% of the refugees under eighteen years old. The available literature indicates that diseases of the skin [6]; digestive [7], respiratory [8] and circulatory systems [9,10]; parasitic diseases [11]; mental disorders [12,13]; and malnutrition [14,15,16,17,18,19] are commonly present among Syrian refugees. Currently, little is known about the impact of prolonged displacement and life in refugee camps on the health of children, who are the most vulnerable group.

Grammatikopoulou et al. [17] observed underweight and stunting in 7.8% and 7.3%, respectively, of children from reception centers in Greece; Pernitez-Agan et al. [18] detected a similarly low prevalence of wasting and stunting (<5% and 10%, respectively), and a higher prevalence of overweight or obesity (10.6%) among Syrian refugee children. In fact, several studies have detected malnutrition among refugees at intake [19,20], but the prelavence of this disease is not known in long-staying refugees and should be studied.

Taking into consideration the especially vulnerable nature of refugee children as reflected in the literature [14,15,16,17,18,19], the aim of this study was to evaluate the nutritional status among long-staying child refugees, according to their gender, in four different refugee residence locations, using anthropometric measurements.

## 2. Materials and Methods

This cross-sectional study was carried out during March and April 2017, in three camps: Zalhé (eastern Lebanon), Deddeh (northern Lebanon), and Kfar Jouz (southern Lebanon), and in the town of Yohmor (southern Lebanon) (Figure 1). The ©Flourish (https://flourish.studio (accessed on 9 January 2023)) software, as a library to create visualizations, was used to map the distribution of places selected in ©OpenStreetMap contributors (https://www.openstreetmap.org (accessed on 9 January 2023)). The sample size was calculated with the sample size calculator website: http://www.raosoft (accessed on 9 January 2023), based on an error margin of 5% and a 90% confidence level, resulting in approximately 177 children in the four places. Children, from 1.2 to 16.4 years old, were selected through convenience sampling (also known as haphazard sampling) of families from a comprehensive resident list provided by the refugee camp authorities (United Nations High Commissioner for Refugees (UNHCR)). All parents who had children under the age of sixteen and who had resided in the camps for at least one year were offered participation. The selected family had the option of choosing which of their children, if any or all, would participate in the study.

This study was accepted by the authorities of these locations, approved by the Ethics Committee of the Universitat de Valencia (Spain) (EH-1-2017), and followed the fundamental principles established in the World Medical Association Declaration of Helsinki [21]. Furthermore, the procedure established by the U.S. National Bioethics Advisory Commission [22] and European Commission [23] was used to obtain written parental or guardian consent, while oral consent was obtained from the children when literacy was a barrier for adults.

Anthropometric assessments were conducted during morning hours after a short physical examination; the psychology and developmental stage of the children were considered. For children over 2 years old, body weight was recorded to the nearest 0.1 kg using a digital scale (Model HN 283, Omron Corporation, Shimogyo-ku, Kyoto, Japan), while wearing lightweight clothing and standing erect with head straight and facing forward. For children <2 years old, the mother was weighed alone, and then the mother and child, wearing lightweight clothing, were weighed together. Subsequently, the child’s weight was calculated by the difference between the previous measured values [24].

For children who were unable to stand using an infantometer, height was measured to the nearest 0.1 cm using a wall-mounted stadiometer that was calibrated daily.

Height-for-age Z-score (HAZ), weight-for-age Z-score (WAZ), and weight-for-height (WHZ) Z-score were calculated using the WHO AnthroPlus software (World Health Organization, 2009, Anthro for Personal Computers, Version 3.01, Software for Assessing Growth and Development of the World’s Children), using the WHO child growth standard 2005 version for children aged 0–5 years [25] and the 2007 version for children and adolescents aged 5–19 years [26]. According to the HAZ, WAZ, and WHZ, chronic malnutrition or stunting, global malnutrition or underweight, and acute malnutrition or wasting, also known simply as chronic, global, acute malnutrition (CGAM), was classified as severe if Z-scores were below −3 standard deviations (SDs) and moderate if Z-scores were between −2 SD and −3 SD [25]. The WHZ scores above +2 SD or +3 SD were considered to be children who were overweight or obese, respectively, by the WHO [24].

Furthermore, the BMI-for-age Z-score (BAZ) was used to classify the population according to their nutritional status into the following categories: severe thinness, moderate thinness, normal, overweight, and obese with cut-points of <−3 SD, from ≥−3 SD to <−2 SD, from ≥−2 SD to ≤+1 SD, from >+1 SD to ≤+2 SD, and >+2 SD, respectively [25].

Mid-upper arm circumference (MUAC) [27] was measured on the left arm, at the midpoint of the upper-arm, between the shoulder tip and elbow, and was classified as severe acute malnutrition (SAM), moderate acute malnutrition (MAM), or normal using the cut-points of <11.5 cm, 11.5–12.5 cm, and >12.5 cm, respectively, for children aged 1–5 years [27,28].

Variables were categorized and described using frequency distributions or percentage, mean, and SD. The normality of distribution was assessed through the Kolmogorov–Smirnov test. A statistical analysis for categorical and continuous variables was conducted through a chi-square (χ^2^) test with Yates correction and type III analysis of variance (ANOVA test, *p* < 0.05), respectively, taking into account that the independent variable was the “location” (Zalhé, Deddeh, Kfar Jouz or Yohmor). Significance was set at *p* < 0.05, and two-sided tests were performed. The prevalence of malnutrition in each location, according to gender and age group, was assessed with the chi-square (χ^2^) test or Fisher’s exact test. The statistical analysis was performed using the “R” statistical software (R Foundation for Statistical Computing) version 2.5.1 (2007).

## 3. Results

Sociodemographic characteristics of the studied populations are shown in Table 1. A total of 176 children were included in the study. Children were aged from 1.2 to 16.4 years old (15 to 197 months old) with 62.5% boys and 37.5% girls. The children had lived for 24.5 ± 3.8 months in the same location. All studied children were Arab, Arabic-speaking, Sunni Muslims and were attending school. Furthermore, children lived in households with from 7 to 10 people. The parents did not have specific jobs within the camps and were dedicated to general upkeep of the refugee camps and raising some animals such as sheep or goats.

The HAZ, WAZ, and WHZ of the studied population are shown in the Figure 2 box plots. The highest median values were obtained in Deddeh for WAZ and HAZ, while the highest result for WHZ was in Yohmor. 

Table 2 shows the mean and SD values of the HAZ, WAZ, WHZ, and MUAC for each Syrian refugee’s location, with only the studied HAZ in these children being significant.

According to the HAZ, WAZ, and WHZ values, no CGAM cases were detected in any studied locations (Table 2). According to the MUAC values recorded, no cases of SAM or MAM were detected. There was no significant difference between the locations for anthropometric parameters according to the ANOVA test.

Table 3 shows the values for BAZ (severe thinness, moderate thinness, normal, overweight, and obesity) separated by locations and age groups. Neither severe thinness nor obesity were observed in any location, and overweight was only observed in one individual in Yohmor. The prevalence of moderate thinness was highest in Kfar Jouz followed by in Zalhé, Yohmor, and Deddeh. The Fisher’s exact test showed that there were no significant differences in nutritional status among the age groups.

Table 4 shows the values for BAZ separated by gender.

## 4. Discussion

There is a statistically significant difference in nutritional status by sex. In fact, the prevalence of moderate thinness in females is higher than in males (80.3 vs. 33.0%, respectively). It is important to separate the results by gender because this malnutrition is hidden if we consider the value of the total number of children. Furthermore, the results reflected that, although no children had CGAM, SAM, or MAM, around half of the children were malnourished. This is because they were in the moderately thinness nutritional status category, which can have an effect on their health status. Several studies have shown that there is an increased risk of malnutrition and malnutrition of infants and children during migrations and war [29,30]. However, the absence of acute malnutrition in our study could be explained by UNICEF-supported food distribution and the general distribution of food stamps by the World Food Programme (WFP) [31,32,33]. In our study, women were moderately thinner than men. This situation should be studied in more detail because they are more vulnerable to malnutrition, as demonstrated by several authors [17,34].

Hossain et al. [31] demonstrated that the highest prevalence of chronic malnutrition was in the host population in Jordan (10.5%). Global malnutrition has been observed to be 3.7% in four refugee camps in Northern Greece [35] and 7.8% in two refugee centers in northern Greece (Drama and Kavala) [17], respectively.

In hospital admissions located in Tal-Abyad’s pediatric in-patient care/department, among children <1 year old, 13.0% exhibited severe acute malnutrition [35]. Acute malnutrition was detected by Walpole et al. [36] in 3.7% of the population and in 4.6% of the population by Grammatikopoulou et al. [17]. Hossain et al. [31] detected a prevalence of acute malnutrition in Syrian refugees between 0.3 and 4.4%.

Meanwhile, Syrian refugees in Jordan and Lebanon, according to the WHZ values from 0.23 to 0.39, were slightly overweight as compared with the WHO’s standard [28]. Similar results, regarding overweight, were obtained by Bilukha et al. [34] in the Zaatari camp, while Pernitez-Agan et al. [18] observed a high prevalence of overweight or obesity (10.6%), and also a low prevalence of wasting (<5%). For overweight and obesity, 16% were detected in four refugee camps in Northern Greece [36] and observed that 1.9% had a MUAC more than two standard deviations (2SD).

The prevalence of stunting (17.0%) was significantly higher among children living in the Zaatari refugee camp from Jordan as compared with children outside the camp (9.0%) [34]. Grammatikopoulou et al. [17], Pernitez-Agan et al. [18], and Walpole et al. [36], found a prevalence of stunting of 7.3%, 9.1%, and 17.4% respectively, among refugee children.

Bilukha et al. [34] demonstrated that the prevalences of wasting (11.1% in 2001, 10% in 2006 and 2009) and stunting (31.1%, 28.6%, and 27.5% in 2001, 2006, and 2009, respectively) of children in the Syrian Arab Republic were high years before the crisis. In fact, Spiroski and Nikovska [37] observed the combined impact of undernutrition and overweight/obesity, also known as the “double burden of malnutrition”, among a Syrian refugee children population, located in two transit centers in Macedonia. According to our results, Hossain et al. [31] indicated that the nutritional status of Syrian refugee children was comparable with that of the host communities.

In fact, Rizkalla et al. [38] conducted interviews with Syrian refugee mothers who resided in Jordan and demonstrated that the goal of humanitarian organizations was to assist individual and family processes that would improve factors affecting the physical and mental health of refugee children. In addition, it is necessary to reduce nutritional problems by using conditional cash transfer programs and microfinance to camp residents [39]. These strategies have been used in the Palestinian refugee camps in Jordan [40], in the Dabaab refugee camp in Kenya [41], and the displaced persons camps in Somalia [42], among others.

Al Masri et al. [43] conducted a cross-sectional study with Syrian refugees, in Germany, and reported unbalanced malnutrition with a high intake of total fat and saturated fatty acids together with a deficiency of minerals and vitamins. It is interesting that, among refugees hosted in other countries, Sankar and Huffman [44] verified that 80% of households were food insecure among Syrian refugees in Florida (USA), being greater in American rural areas than in urban areas.

Khuri et al. [45] indicated that, among refugees in host countries, there should be centralized, comprehensive dietary and health screening together with culturally appropriate and sustainable nutritional education tools and interventions for them, which would help to improve their diet, nutrition, and quality of life. For refugees housed in Australia, a high prevalence of vitamin and micronutrient deficiencies has been observed along with overweight/obesity (typical example of double burden of malnutrition) [46]; in Sweden, they had a low health-related quality of life [47]; in Canada, they demonstrated food insecurity [48]. Unfortunately, forced migration to other countries generates health problems due to acculturation factors that affect early and later stages and post-resettlement, giving rise to temporary stressors related to seeking asylum with adaptation to new environments and exclusion social problems with the labor market [49].

In addition, several nutrition-related pathologies have been associated with Syrian refugees, such as (i) anemia due to inadequate iron intake by women during pregnancy and (ii) suboptimal maternal and child nutrition due to increased nutrient requirements [10], diabetes and hypertension [50], and gastrointestinal infections [51], among others. Furthermore, the COVID-19 pandemic affected Syrian refugees who lived outside the camp, as they had more isolation affecting their physical, mental, social, and economic health in Jordan [52]. Affected in other countries were health care and social support, among others, and impediments to border crossing resulting in a combined effect in Canada [53], increased food insecurity in Lebanon [54], biological effects and socioeconomic conditions in Lebanon and Turkey [55], and anxiety in the Al-Zaatari refugee camp (Jordan) [56].

The use of a prediction model of this pandemic in Syrian refugees could be useful. McCall et al. [57] conducted a cross-sectional study in Lebanon, and noted that predictors of the inability to manage noncommunicable diseases among these refugees were due to financial barriers. Furthermore, they [57] reflected on the future of these refugees, from a post-COVID-19 point of view, and highlighted the importance of institutional and financial support that should be the impetus for refugee management in their emergency travel and search demands, with synergistic performance within government to manage the crisis.

All this implies that adequate assistance is required to overcome financial barriers and allow equitable access to medicines and medical care to guarantee the health and quality of life of Syrian refugees.

The importance and implications of our study demonstrated that the continuous analysis of anthropometric measurements in this population must be carried out, in addition to other measurements, to ensure the health and successful growth of children.

## 5. Study Limitations

The limitations of this study are that we had a small sample and low female participation, which could impact the interpretation of the results and underestimate the presence of malnutrition. From our point of view, we believe that the reasons for the low female participation rate could be due to protective family attitudes towards girls and not allowing them to participate in the study, together with the fact that participating women preferred doctors of the same gender, geographic location, and cultural background, which has been demonstrated in several studies [58,59,60]. In addition, a possible solution to this situation could be the incorporation of women health providers and interpreters into medical teams, such as was reflected in the literature [61,62]. Despite this, this study provided useful information for assessing nutritional status using anthropometric tools. However, it should be noted that, in order to correctly interpret the results of other published studies and to compare the results, it is key to identify which child growth standard is used, i.e., Bucak et al. [19] used the Waterlow and Gomez classification, but other studies determined nutritional status based on WHO’s standardized approach.

## 6. Conclusions

In conclusion, the present data demonstrate that the prevalence of malnutrition appears low among long-staying refugee children. However, when interpreting the data, one must consider that their stay in the camps for more than two years has not protected them against malnutrition, which was not expected to be found. Given that a significant number of refugees are children, more research is needed on the health status of this population. There is also a need to evaluate the actual objective strategies for assessing the health of children refugees. The main objective for this population should be to apply safety nets, interventions, and programs to address this issue.

## Figures and Tables

**Figure 1 life-13-00453-f001:**
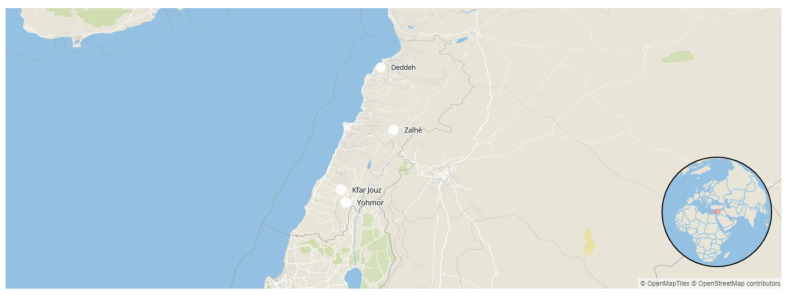
Localization of the refugee camps studied using ©Flourish studio and ©OpenStreetMap.

**Figure 2 life-13-00453-f002:**
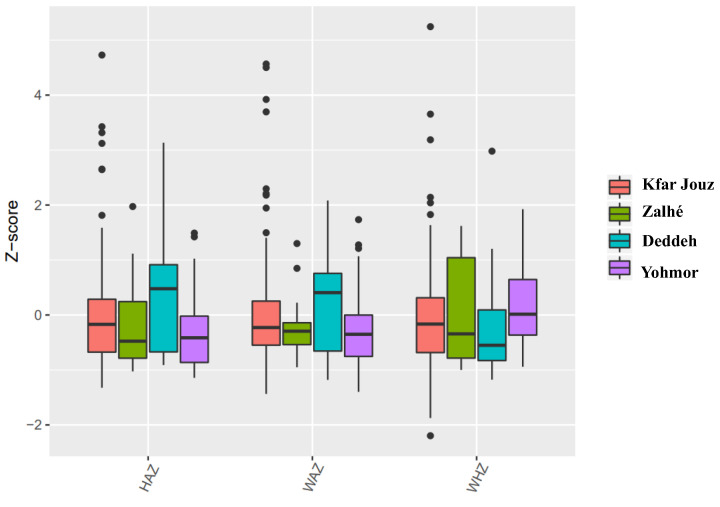
Normalized data of the WAZ, HAZ, and WHZ, for each location, showing the median, maximum, and minimum values.

**Table 1 life-13-00453-t001:** Sociodemographic characteristics of studied children (n = 176).

Variable	n (%)	*p*-Value
Gender		
Male	110 (62.5)	0.003 *
Female	66 (37.5)	
Age		
15–23 months	3 (1.7)	0.001 *
24–35 months	11 (6.3)	
36–47 months	20 (11.4)	
48–59 months	27 (15.3)	
5–6 years	19 (10.8)	
6–7 years	17 (9.7)	
7–8 years	18 (10.2)	
8–9 years	9 (5.1)	
9–10 years	19 (10.8)	
10–11 years	12 (6.8)	
11–12 years	9 (5.1)	
12–13 years	7 (4.0)	
13–14 years	2 (1.1)	
14–15 years	2 (1.1)	
15–16 years	1 (0.6)	
Age (years)		
<5 years	64 (36.4)	
≥5–<10 years	79 (44.9)	
≥10–14 years	33 (18.8)	
Male	87.35 ± 38.92 months	0.664 **
Female	90.52 ± 36.11 months	

* Chi square test, ** ANOVA.

**Table 2 life-13-00453-t002:** Anthropometric values by location for the studied children.

Location		HAZ	WAZ	WHZ	MUAC
Yohmor	Mean	1.4798	0.1933	0.1933	15.291
(n = 102)	Standard deviation	0.49107	0.03374	0.03374	1.862
Kfar Jouz	Mean	1.5719	0.1917	0.1917	18.24
(n = 15)	Standard deviation	0.63156	0.04606	0.04606	2.838
Deddeh	Mean	1.9076	0.1746	0.1746	17.169
(n = 23)	Standard deviation	0.72461	0.04482	0.04482	2.525
Zalhé	Mean	1.4798	0.1933	0.1933	17.496
(n = 36)	Standard deviation	0.49107	0.03374	0.03374	2.373
Total	Mean	1.5818	0.1892	0.1892	17.679
(n = 176)	Standard deviation	0.58426	0.03829	0.03829	1.862
*p*-value *		0.023	0.223	0.238	0.174

* ANOVA test; HAZ, height-for-age Z-score; WAZ, weight-for-age Z-score; WHZ, weight-for-height Z-score; MUAC, mid-upper arm circumference.

**Table 3 life-13-00453-t003:** Prevalence (%) of nutritional status according to the BAZ by location and age.

Place	Nutritional Status According to BAZ	<5 Years Old	≥5–<10 Years Old	≥10–14 Years Old	Total Sample
		n (%)	n (%)	n (%)	n (%)
Yohmor (n = 102)	Severe thinness	-	-	-	-
	Moderate thinness	19 (48.7)	27 (58.7)	5 (29.4)	51 (50.0)
	Normal	20 (51.3)	19 (41.3)	11(64.7)	50 (49.0)
	Overweight	-	-	1 (5.9)	1 (1.0)
	Obesity	-	-	-	-
	Total	39 (100.0)	46 (100.0)	17 (100.0)	102 (100.0)
Kfar Jouz (n = 15)	Severe thinness	-	-	-	-
	Moderate thinness	4 (80.0)	3 (60.0)	2 (40.0)	9 (60.0)
	Normal	1 (20.0)	2 (40.0)	3 (60.0)	6 (40.0)
	Overweight	-	-	-	-
	Obesity	-	-	-	-
	Total	5 (100.0)	5 (100.0)	5 (100.0)	15 (100.0)
Deddeh (n = 23)	Severe thinness	-	-	-	-
	Moderate thinness	5 (38.5)	1 (14.3)	2 (66.7)	8 (34.8)
	Normal	8 (61.5)	6 (85.7)	1 (33.3)	15 (65.2)
	Overweight	-	-	-	-
	Obesity	-	-	-	-
	Total	13 (100.0)	7 (100.0)	3 (100.0)	23 (100.0)
Zalhé (n = 36)	Severe thinness	-	-	-	-
	Moderate thinness	3 (42.9)	12 (57.1)	4 (50.0)	19(52.8)
	Normal	4 (57.1)	9 (42.9)	4 (50.0)	17 (47.2)
	Overweight	-	-	-	-
	Obesity	-	-	-	-
	Total	7 (100.0)	21 (100.0)	8 (100.0)	36 (100.0)
Total (n = 176)	Severe thinness	-	-	-	-
	Moderate thinness	31 (48.4)	43 (54.4)	13 (39.4)	87 (49.4)
	Normal	33 (51.6)	36 (45.6)	19 (57.6)	88 (50.0)
	Overweight	-	-	1 (3.0)	1(0.6)
	Obesity	-	-	-	-
	Total	64 (100.0)	79 (100.0)	33 (100.0)	176 (100.0)
*p*-value *		0.457	0.171	0.845	0.732

* Fisher’s exact test; BAZ, body mass index-for-age Z-score.

**Table 4 life-13-00453-t004:** Prevalence (%) of nutritional status according to the BAZ by gender.

Nutritional Status According to BAZ	Female n (%)	Male n (%)	Total n (%)
Severe thinness	-	-	-
Moderate thinness	49 (80.3)	38 (33.0)	87 (49.4)
Normal	12 (19.7)	76 (66.1)	88 (50.0)
Overweight	-	1 (0.9)	1 (0.6)
Obesity	-	-	-
Total	61 (100.0)	115 (100.0)	176 (100.0)
*p*-value *	<0.05	<0.05	>0.05

* Fisher’s exact test; BAZ, body mass index-for-age Z-score.

## Data Availability

The data presented in this study are available on request from the corresponding author.

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
