# Peer review of "Prevalence of Malnutrition among Syrian Refugee Children from Lebanon"

_life, 2023, doi:10.3390/life13020453_

Round 1

Reviewer 1 Report

It is an interesting article regarding the malnutrition in chidren in Syrian refugee camps. I have a few comments:

- in the abstract: 176 Syrian refugee children, with stays of more than two years at three refugee camps... and at the results there are 3 children under 2 years

- materials and methods you said that oral consent was given... at the end you said Written parental consent and verbal consent from the child had been obtained to publish this paper.

- also the article is too short you should improve and extend the discussions

Author Response

Reviewer’s comment: It is an interesting article regarding the malnutrition in chidren in Syrian refugee camps.

Author’s comment:  Thank you for your comment.

Reviewer’s comment: in the abstract: 176 Syrian refugee children, with stays of more than two years at three refugee camps... and at the results there are 3 children under 2 years

Author’s comment: Sorry for this mistake. The reference of 2 years is reflected in the materials and methods section where is indicated that “Body weight, was recorded, to the nearest 0.1 kg, using a digital scale (Model HN 283, Omron Corporation, Shimogyo-ku, Kyoto, Japan) while wearing lightweight clothing and standing erect with heads straight facing forward for children more than 2 years old. For children aged <2 years old, the mother was weighed alone and then the mother and child wearing lightweight clothing were weighed together”.  We have changed two years in this phrase “All parents who had children under the age of 16 and who had resided in the camps for at least ONE year were offered participation” due to those three children is under tow years.

Reviewer’s comment: materials and methods you said that oral consent was given... at the end you said Written parental consent and verbal consent from the child had been obtained to publish this paper.

Author’s comment: It is changed in the manuscript due to that the procedure is the following: “Written parental or guardians’ consents were obtained, while oral consent was carried out from the child or when literacy is a barrier for adults” being based in the U.S. National Bioethics Advisory Commission, World Medical Association and European Commission.

Reviewer’s comment: Also, the article is too short you should improve and extend the discussions

Author’s comment: According to your comment, it has been carried out.

Reviewer 2 Report

The topic is very interesting and important. However, the manuscript can be greatly improved as mentioned below.

Major comments:

-        Given that this is a refugee study, it is important to mention in which country is has been conducted in the title and abstract.

-        Since this is a study conducted in Lebanon, it is important to bring up recent literature available on anemia and nutritional status among Syrian refugee mothers and children under five in Lebanon (pubmed: 34199032 and 36364722) in the introduction and discussion. These studies could also be used in the discussion as wasting, stunting, and underweight were detected.

-        Introduction/objective (Line 63): the authors do not justify enough why this study is important among refugee children. In addition, authors need to indicate whether there is a gap in the literature and mention existing studies or surveys conducted in Lebanon.

-        No sample size calculation was provided. Please explain and add justification.

-        I find it odd that no cases of stunting or wasting or underweight were found, especially that moderate thinness was recorded in Table 3. I suggest that a breakdown is presented in a table, in addition to the data presented in Table 2. It is not enough to show the mean, it would valuable to show n(%) according to the classification of severe and moderate.

-        Discussion (Line 170): There is a high chance that as a good proportion of the children included in the study were born after the war started and mostly likely in the refugee camps in Lebanon and another good proportion mostly likely arrived to Lebanon when they were very young… so the explanation that a pre-crisis nutritional status was better is not very strong, unless you can provide more details on the studied age groups in the cited references.

-        Line 170-171: “the adaptability of the human being to a long-term conflict” does not have a supporting reference. Many studies and reports show that there is a higher risk for malnutrition and poor infant and child nutrition during migrations and war. I suggest you elaborate on this argument and justify it.

Minor comments:

-        Line 18-20: “this study…refugees” is not very clear. Please reformulate the sentence.

-        Figure 1 is blurred and the name of the cities is not visible enough. Please provide a higher resolution.

-        Methods (Line 69): Please explain the purpose of Flourish and why it was used. It is not well explained.

-        Line 73: please clarify “camp authorities”. Is it UNHCR? Or is it another entity?

-        Line 74: what is the age group studied? From 0 to 16 years old? It is not clear.

-        Line 75: please justify why multiple children were selected from the same household. It is common that one child is selected from one household as they share the same socio-economic characteristics.

-        Line 84: “assent was sought from the children when possible”, was there an age group that could do it? Please specify.

-        Lines 93-97: it is not very clear how height was measured. Please reformulate for clarity. Please indicate clearly for which age groups the height or the length was measured.

-        Table 1: since the study sample have children up to the age of 16 years, it makes sense to also show that in the table instead of showing children aged 60 months or older (65%). It makes sense to add more age brackets instead of having the majority of children fall in the last category (that was later shown in Table 3, please add it to table 1 too).

-        Figure 2: it would be nice to add a sentence describing the findings in the body on the manuscript.

-        Table 2: please add Table 2 after its first mention in-text.

-        Line 146-149: these findings are based on figure 2 or table 2 or both? Please specify.

-        Line 160-162: keep this idea for the discussion instead of the results’ section.

-        Table 4: it would be interesting to show wasting, stunting, and underweight according to gender.

-        Line 183: I wouldn’t generalize “all”. It would be better to provide a percentage.

-        Line 209: there is an extra “t” at the end.

Line 228-229: it is too general. You can suggest that safety nets, interventions, and programs could be in place to address this issue

Author Response

Reviewer’s comment: The topic is very interesting and important. However, the manuscript can be greatly improved as mentioned below.

Author’s comment: According to your comments, we think that the reviewed manuscript is improved.

Reviewer’s comment: Major comments:

-        Given that this is a refugee study, it is important to mention in which country is has been conducted in the title and abstract.

Author’s comment: It has modified in the title and abstract.

Reviewer’s comment: Since this is a study conducted in Lebanon, it is important to bring up recent literature available on anemia and nutritional status among Syrian refugee mothers and children under five in Lebanon (pubmed: 34199032 and 36364722) in the introduction and discussion. These studies could also be used in the discussion as wasting, stunting, and underweight were detected.

Author’s comment: According to your comment, both references have added in the introduction and discussion sections.

Reviewer’s comment: Introduction/objective (Line 63): the authors do not justify enough why this study is important among refugee children. In addition, authors need to indicate whether there is a gap in the literature and mention existing studies or surveys conducted in Lebanon.

Author’s comment: According to your comment, we have added this idea in that paragraph.

Reviewer’s comment: No sample size calculation was provided. Please explain and add justification.

Author’s comment: It has been explained and justified in the manuscript.

Reviewer’s comment: I find it odd that no cases of stunting or wasting or underweight were found, especially that moderate thinness was recorded in Table 3. I suggest that a breakdown is presented in a table, in addition to the data presented in Table 2. It is not enough to show the mean, it would valuable to show n(%) according to the classification of severe and moderate.

Author’s comment: In our viewpoint, Tables 2 and 3 helped to demonstrate values. Table 2 focussed in the values HAZ, WAZ and WHZ, no CGAM cases were detected in any studied location (Table 2). According to the MUAC values recorded, no cases of SAM or MAM were detected. There was no significant difference between the locations for anthropometric parameters according to the ANOVA test. However, we think that table 3 shows the values for BAZ (severe thinness, moderate thinness, normal, over-weight and obesity) separated by locations and age groups. Neither severe thinness nor obesity were observed in any location and overweight was only observed in one individu-al in Yohmor. The prevalence of moderate thinness is highest in Kfar Jouz followed by Zalhé, Yohmor and Deddeh. The Fisher's exact test showed that there were no significant differences in nutritional status among the age groups.

Reviewer’s comment: Discussion (Line 170): There is a high chance that as a good proportion of the children included in the study were born after the war started and mostly likely in the refugee camps in Lebanon and another good proportion mostly likely arrived to Lebanon when they were very young… so the explanation that a pre-crisis nutritional status was better is not very strong, unless you can provide more details on the studied age groups in the cited references.

Author’s comment: According to this comment, this idea has been deleted in the discussion section.

Reviewer’s comment: Line 170-171: “the adaptability of the human being to a long-term conflict” does not have a supporting reference. Many studies and reports show that there is a higher risk for malnutrition and poor infant and child nutrition during migrations and war. I suggest you elaborate on this argument and justify it.

Author’s comment: According to this comment, we have re-written to clarify it.

Reviewer’s comment: Minor comments:

- Line 18-20: “this study…refugees” is not very clear. Please reformulate the sentence.

Author’s comment:

Reviewer’s comment: Figure 1 is blurred and the name of the cities is not visible enough. Please provide a higher resolution.

Author’s comment: According to your comment, Figure 1 has been improved and included in the manuscript.

Reviewer’s comment: Methods (Line 69): Please explain the purpose of Flourish and why it was used. It is not well explained.

Author’s comment: The Flourish is useful to prepare and share with the community visualizations. The use of Flourish is reflected in several studies in MDPI:

  • Bernasconi, A., & Grandi, S. (2021). A conceptual model for geo-online exploratory data visualization: The case of the covid-19 pandemic. Information, 12(2), 69.
  • Uršulin-Trstenjak, N., Dodlek Šarkanj, I., Sajko, M., Vitez, D., & Živoder, I. (2021). Determination of the Personal Nutritional Status of Elderly Populations Based on Basic Foodomics Elements. Foods, 10(10), 2391.
  • Kovac, T., Sarkanj, B., Borisev, I., Djordjevic, A., Jovic, D., Loncaric, A., ... & Krska, R. (2020). Fullerol [C. sub. 60][(OH). sub. 24] Nanoparticles Affect Secondary Metabolite Profile of Important Foodborne Mycotoxigenic Fungi In Vitro. Toxins, 10(4), 1g-1g.

The explanation of its use is added in the material and methods section

Reviewer’s comment: Line 73: please clarify “camp authorities”. Is it UNHCR? Or is it another entity?

Author’s comment: It is United Nations High Commissioner for Refugees (UNHCR) being added in the manuscript.

Reviewer’s comment: Line 74: what is the age group studied? From 0 to 16 years old? It is not clear.

Author’s comment: Children were aged from 1.2 to 16.4 years old. It has been added in the manuscript.

Reviewer’s comment: Line 75: please justify why multiple children were selected from the same household. It is common that one child is selected from one household as they share the same socio-economic characteristics.

Author’s comment: In our viewpoint, it is common that one child is selected from one household. However, we carried out the study using multiple children from the same household because as is observed in our results the prevalence of moderate thinness in females is higher than males. It could indicate that children not received the same type of ration. It is applied in other studies as are:

  • Jamaluddine, Z., Sahyoun, N. R., Choufani, J., Sassine, A. J., & Ghattas, H. (2019). Child-reported food insecurity is negatively associated with household food security, socioeconomic status, diet diversity, and school performance among children attending UN Relief and Works Agency for Palestine Refugees schools in Lebanon. The Journal of nutrition, 149(12), 2228-2235.
  • Mackintosh, U. A. T., Marsh, D. R., & Schroeder, D. G. (2002). Sustained positive deviant child care practices and their effects on child growth in Viet Nam. Food and nutrition bulletin, 23(4_suppl_1), 16-25.

Reviewer’s comment: Line 84: “assent was sought from the children when possible”, was there an age group that could do it? Please specify.

Author’s comment: It has been re-written in the manuscript because is confusing. In fact, we worked with the procedure established by U.S. National Bioethics Advisory Commission and European Commission was used to obtain written parental or guardians’ consents, while oral consent was carried out from the child or when literacy is a barrier for adults.

Reviewer’s comment: Lines 93-97: it is not very clear how height was measured. Please reformulate for clarity. Please indicate clearly for which age groups the height or the length was measured

Author’s comment: According to your comment, we have re-written this paragraph to clarify it.

Reviewer’s comment: Table 1: since the study sample have children up to the age of 16 years, it makes sense to also show that in the table instead of showing children aged 60 months or older (65%). It makes sense to add more age brackets instead of having the majority of children fall in the last category (that was later shown in Table 3, please add it to table 1 too).

Author’s comment: According to your comment, we have modified Table 1.

Reviewer’s comment: Figure 2: it would be nice to add a sentence describing the findings in the body on the manuscript.

Author’s comment: According to your comment, it has added in the manuscript.

Reviewer’s comment: Table 2: please add Table 2 after its first mention in-text.

Author’s comment: According to your comment, it has added in the manuscript.

Reviewer’s comment: Line 146-149: these findings are based on figure 2 or table 2 or both? Please specify.

Author’s comment: These findings are based on Table 2. It has added in the manuscript.

Reviewer’s comment: Line 160-162: keep this idea for the discussion instead of the results’ section.

Author’s comment: According to your comment, it has been added in the discussion section.

Reviewer’s comment: Table 4: it would be interesting to show wasting, stunting, and underweight according to gender.

Author’s comment: Wasting, stunting and underweight is reflected as  WHZ, HAZ and WAZ, respectively, being these parameters indicated in previous tables and figure. Table 4 reflected the value of BAZ by gender, being BAZ classified by severe thinness, moderate thinness, normal, overweight and obesity. According to WHO, WHZ, HAZ, WAZ and BAZ are different parameters.

According to the HAZ, WAZ and WHZ, chronic malnutrition or stunting, global malnu-trition or underweight and acute malnutrition or wasting,

Reviewer’s comment: Line 183: I wouldn’t generalize “all”. It would be better to provide a percentage.

Author’s comment: We have re-written to clarify it.

Reviewer’s comment: Line 209: there is an extra “t” at the end.

Author’s comment: According to your comment, it has been deleted.

Reviewer’s comment: Line 228-229: it is too general. You can suggest that safety nets, interventions, and programs could be in place to address this issue

Author’s comment: According to your comment, it has been added.

Reviewer 3 Report

The aim of this study was to evaluate the nutritional status among long-term child refugees in four different refugee residence locations using anthropometric measurements in camps: Zalhé (eastern Lebanon), Deddeh (northern Lebanon) and Kfar Jouz (southern Lebanon), and in Yohmor town (southern Lebanon). In their results, the authors identified that the prevalence of malnutrition appears low among long-term refugee children. In the introduction part: -   I have no comments, all the cited references are relevant to the research. In the material and methods section: - in the title of Figure 1. Localization of the refugee camps studied. (Line 78), it would be appropriate to add the source for the location map in parentheses (maybe it was the used software? - I especially appreciate the ethical aspect of performing anthropometric measurements, considering the psychology and developmental stage of the children being measured. Results I have no comments, the results are presented clearly, at an appropriate scientific level, and considering the fulfillment of the study's goal, I consider them sufficient. Discussion -   I have no comments. Conclusion - In the conclusion, it would be appropriate to add in line 223 a specific figure for the prevalence of malnutrition among children, determined from the results of this study. References - 37 literary sources were used in the manuscript (without self-citations), of which 25 sources are from the last 5 years; 6 sources for the last 5-10 years and 6 sources that are older than 10 years (but in this case they form 4 sources of classification and WHO standards for assessing the state of nutrition or malnutrition, therefore these sources are fully justified).

Author Response

Reviewer’s comment: The aim of this study was to evaluate the nutritional status among long-term child refugees in four different refugee residence locations using anthropometric measurements in camps: Zalhé (eastern Lebanon), Deddeh (northern Lebanon) and Kfar Jouz (southern Lebanon), and in Yohmor town (southern Lebanon). In their results, the authors identified that the prevalence of malnutrition appears low among long-term refugee children. In the introduction part: -   I have no comments, all the cited references are relevant to the research. 

Author’s comment: Thank you for your comment.

Reviewer’s comment: In the material and methods section: - in the title of Figure 1. Localization of the refugee camps studied. (Line 78), it would be appropriate to add the source for the location map in parentheses (maybe it was the used software? - I especially appreciate the ethical aspect of performing anthropometric measurements, considering the psychology and developmental stage of the children being measured. 

Author’s comment: ©Flourish (https://flourish.studio) software, as library to create visualizations, was used to map the distribution of places selected in ©OpenStreetMap contributors (https://www.openstreetmap.org). It has been added in the title of Figure and re-written in materials and methods section. Furthermore, your comment about ethical aspect has been added in the manuscript.

Reviewer’s comment: Results I have no comments, the results are presented clearly, at an appropriate scientific level, and considering the fulfillment of the study's goal, I consider them sufficient. 

Author’s comment: Thank you for your comment.

Reviewer’s comment: Discussion -   I have no comments. 

Author’s comment: Thank you for your comment.

Reviewer’s comment: Conclusion - In the conclusion, it would be appropriate to add in line 223 a specific figure for the prevalence of malnutrition among children, determined from the results of this study. 

Author’s comment: We think that add this specific figure duplicate results reflected in the manuscript.

Reviewer’s comment: References - 37 literary sources were used in the manuscript (without self-citations), of which 25 sources are from the last 5 years; 6 sources for the last 5-10 years and 6 sources that are older than 10 years (but in this case they form 4 sources of classification and WHO standards for assessing the state of nutrition or malnutrition, therefore these sources are fully justified).

Author’s comment: Thank you for your comment. We have added some references to justify reviewers’ comments.

Round 2

Reviewer 1 Report

The article has been improved a lot, but still at the end of the article is missing the Institutional Review Board Statement:

Author Response

Comments and Suggestions for Authors: The article has been improved a lot, but still at the end of the article is missing the Institutional Review Board Statement.

Author’s comment: According to your comment, we have added this section.

Reviewer 2 Report

Thank you for addressing the comments adequately, the manuscript has improved. However, it needs to be checked by an English native speaker as some sentences are too long and others may have grammatical errors, especially in the discussion.

Minor comment:

-        the first sentence of the introduction is unclear.

-        Thank you adding the age brackets in table 1, but it is not practical to read the age in months until the age of 16 years without holding a calculator. I think it is fine to use months for the first 5 years and then switch to years as long as the use of the unit is clear.

Author Response

Comments and Suggestions for Authors: Thank you for addressing the comments adequately, the manuscript has improved. However, it needs to be checked by an English native speaker as some sentences are too long and others may have grammatical errors, especially in the discussion.

Author’s comment: According to your comment, an English native speaker has reviewed this morning the manuscript. We have modified according to his comments.

Reviewer’s comment: Minor comment:

-        the first sentence of the introduction is unclear.

Author’s comment: According to your comment, we have modified this first sentence

Reviewer’s comment:  Thank you adding the age brackets in table 1, but it is not practical to read the age in months until the age of 16 years without holding a calculator. I think it is fine to use months for the first 5 years and then switch to years as long as the use of the unit is clear.

Author’s comment: According to your comment, we have modified this idea in table 1.